# A LANGUAGE MODEL BASED MODEL MANAGER

## ABSTRACT

In the current landscape of machine learning, we face a "model lake" phenomenon: a proliferation of deployed models often lacking adequate documentation. This presents significant challenges for model users attempting to navigate, differentiate, and select appropriate models for their needs. To address the issue of differentiation, we introduce Model Manager, a framework designed to facilitate easy comparison among existing models. Our approach leverages a large language model (LLM) to generate verbalizations of two models' differences by sampling from two models. We use a novel protocol that makes it possible to quantify the informativeness of the verbalizations. We also assemble a suite with a diverse set of commonly-used models: Logistic Regression, Decision Trees, and K-Nearest Neighbors. We additionally performed ablation studies on crucial design decisions of the Model Managers. Our analysis yields pronounced results. For a pair of logistic regression models with a 20-25% performance difference on the blood dataset, the Model Manager effectively verbalizes their variations with up to 80% accuracy. The Model Manager framework opens up new research avenues for improving the transparency and comparability of machine learning models in a post-hoc manner.

## 1 INTRODUCTION

The rapid increase in the number of machine learning models across various domains has led to the saturation of these models, many of which are poorly documented and lack standardized evaluation metrics. This abundance creates a "model lake" (Pal et al., 2024), a vast and complex landscape where navigating and selecting models for specific tasks is increasingly challenging since it's often a struggle to discern the strengths and weaknesses of these models. Several efforts have been made to improve model management and documentation. One example is ModelDB (Vartak et al., 2016), which serves as a versioning system that tracks models' metadata across successive iterations (such as model configurations, training datasets, and evaluation metrics). ModelDB's primary focus is on ensuring reproducibility and traceability of models over time, allowing users to track changes and reproduce past experiments. Similarly, Model Cards (Mitchell et al., 2019) and Data Cards (Pushkarna et al., 2022), along with recent work on their automated generation (Liu et al., 2024), offer valuable documentation on data characteristics, model architectures, and training processes. While these methods provide critical insights into individual models and datasets, they do not explicitly dive into verbalizing the differences in model predictions across the feature space. Addressing these limitations and providing interpretable verbalizations is essential for enabling more informed decisions when selecting or developing new and effective models. Yet, research aimed at systematically differentiating models remains sparse, leaving room for innovation in model transparency and comparison techniques.

Recently, Large Language Models (LLMs) have shown exceptional capabilities over a diverse range of tasks (Hendy et al., 2023; Brown et al., 2020). Previous work has shown that LLMs can be leveraged to explain model behavior (Kroeger et al., 2023) and to develop explanation methods for other modules (Singh et al., 2023). These advancements motivate us to build a "Model Manager" framework that leverages LLMs to verbalize the model differences.

The Model Manager framework is designed to compare two models trained on the same dataset by capturing and verbalizing their differences. It does so by serializing a representative sample of input instances (from the dataset) and the corresponding model outputs in a JSON format. The serialization, along with a task description, is passed to the LLM through a zero-shot-based prompt. The LLM then

analyzes the patterns from the serialization, captures the inconsistencies in the predictions between the two models, and summarizes them in human-understandable texts.

The Model Manager framework is flexible. Since the framework primarily relies on comparing input-output samples, it can be used with various model types and datasets. Additionally, the Model Manager is extensible. The framework allows the user to incorporate model-specific information, for example, textual descriptions of the structures of decision trees, which can improve the informativeness of the verbalization — we present the effects via ablation studies in Section 6.

To evaluate the verbalization of Model Manager, we set up a novel protocol that is inspired by the evaluation of natural language explanations (Kopf et al., 2024; Singh et al., 2023). Given the inputs, the first model's outputs, and the verbalization, we use an external LLM to infer the second model's output. The accuracy of the inference is used to quantify the quality of the verbalization.

We test and compare the Model Managers utilizing state-of-the-art LLMs through a series of experiments across different datasets, and model types (Logistic Regression, Decision Tree, K-Nearest Neighbor). Our investigation reveals the following key findings:

- The framework can effectively verbalize differences between model-based learning algorithms.

- Providing access to models' internals (e.g., learned parameters) leads to more accurate verbalizations.

- Obfuscating model-type information from our framework has no statistically significant effect on its performance.

We demonstrate that our work provides a valuable starting point for future directions in explainable artificial intelligence (XAI) where LLMs can be used to manage models and enhance their transparency and comparability in a post-hoc manner.

## 2 RELATED WORKS

**Neuron-Level Semantics**   Research into the semantics of individual DNN components, particularly neurons, has evolved significantly. Early investigations, such as those by Mu and Andreas (2020), focused on identifying compositional logical concepts within neurons. Building on this, Hernandez et al. (2022) developed techniques to map textual descriptions to neurons by optimizing pointwise mutual information. More recent approaches have incorporated external models to enhance explanations of neuron functions. For instance, Bills et al. (2023) conducted a proof-of-concept study using an external large language model (LLM), such as GPT-4, to articulate neuron functionalities. However, the perfection of these methods remains elusive, as noted by Huang et al. (2023). Evaluating the effectiveness of these explanations is currently a vibrant area of inquiry, with ongoing studies like those by Kopf et al. (2024) and Mondal et al. (2024).

**Model-Level Explanations**   Beyond individual neurons, the field is extending towards automated explanation methods for broader model components. Singh et al. (2023) approaches models as opaque "text modules," providing explanations without internal visibility. Our methodology diverges by incorporating more detailed information about the models, which we believe enhances the accuracy of explanations, a concept supported by Ajwani et al. (2024). Notably, our work aligns with Kroeger et al. (2023), who employ in-context learning for prompting LLMs to explain machine learning models. Our strategy differs as we emphasize zero-shot instructions.

**Interpretable Feature Extraction**   Concurrently, there is a shift towards extracting interpretable features directly from neurons. Techniques such as learning sparse auto-encoders have been explored by Bricken et al. (2023). A significant advancement by Templeton et al. (2024) scales up these efforts to newer architectures like Claude 3.5 Sonnet (Anthropic, 2024). Unlike previous methods, we do not assume a predefined set of features for explanation, opting instead to use the LLM as a dynamic "model manager" to generate explanatory content.

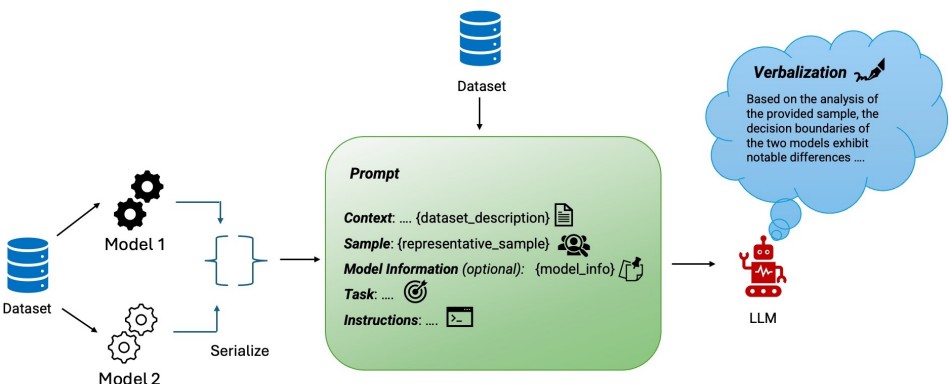

Figure 1: **Overview of the model manager framework**: Given a dataset and a pair of models trained on that dataset, the framework verbalizes the differences between the two models.

**Verbalization Techniques**    Another prevalent approach is the use of the language model head of DNNs as a "logit lens," as demonstrated by nostalgebraist (2020). This method has been further developed and diversified by researchers like Pal et al. (2023) and Belrose et al. (2023). The PatchScope framework by Ghandeharioun et al. (2024) extends these techniques, incorporating methods that modify the representations themselves. In our research, rather than utilizing the language model head directly, we employ an external LLM to serve as the "model manager," providing a novel means of interpreting and explaining model behaviors.

**LLM Distinction**    Several approaches have emerged to differentiate between LLMs. One method, LLM Fingerprinting, introduces a cryptographically inspired technique called Chain and Hash (Russinovich and Salem, 2024). This approach generates a set of unique questions (the "fingerprints") and corresponding answers, which are hashed to prevent false claims of ownership over models. Complementing this, another method (Richardeau et al., 2024) proposes using a sequence of binary questions, inspired by the 20 Questions game, to determine if two LLMs are identical. Unlike fingerprinting or binary distinction, our framework focuses on the behavioral aspect of models. Moreover, our current work does not aim to compare LLMs themselves; rather, we leverage LLMs as a tool to compare and verbalize the differences among other models.

## 3 THE MODEL MANAGER

Here we present our framework (as illustrated by Figure 1) that generates natural-language descriptions of the differences between two ML models trained on the same dataset, i.e., the verbalizations.

**Notation:**    Let $\mathbf{X} = \{\mathbf{x_i}\}_{i=1}^{n}$ be a tabular dataset where each $\mathbf{x_i} \in \mathbb{R}^d$ represents a feature vector. Since we consider classification, suppose the target vector is $\mathbf{y} = \{y_i\}_{i=1}^{n}$, where $y_i \in C$ and $C$ is a set of possible classes. We denote a subset of the dataset as $\mathbf{X}_{\text{sub}}$, with size $n_{\text{sub}}$. Similarly, the corresponding subset of target values is denoted by $\mathbf{y}_{\text{sub}} = \{y_i\}_{i=1}^{n_{\text{sub}}}$. We define the feature names of $\mathbf{X}$ as $F = \{f_1, f_2, \ldots, f_d\}$, where each $f_i$ represents a natural-language description of a feature, such as "age" or "glucose."

Let $M_1$ and $M_2$ be the two models that we compare with our Model Manager. For each data point $\mathbf{x_i} \in \mathbf{X}_{\text{sub}}$, the predicted target values from models $M_1$ and $M_2$ are represented as $\hat{y}_{\text{sub},i}^{(1)} = M_1(\mathbf{x_i})$ and $\hat{y}_{\text{sub},i}^{(2)} = M_2(\mathbf{x_i})$, respectively. The corresponding predicted target vectors for the subset are denoted by $\hat{\mathbf{y}}_{\text{sub}}^{(1)}$ and $\hat{\mathbf{y}}_{\text{sub}}^{(2)}$.

**Representative Sample:**    We construct our representative sample using the $verb$ split of the dataset $\mathbf{X}_{\text{verb}}$ (size $n_{\text{verb}}$) along with the predicted target vectors $\hat{\mathbf{y}}_{\text{verb}}^{(1)}$ and $\hat{\mathbf{y}}_{\text{verb}}^{(2)}$ from models $M_1$ and $M_2$

respectively. Before passing the verbalization sample $\{\mathbf{X}_{\text{verb}}, \hat{\mathbf{y}}_{\text{verb}}^{(1)}, \hat{\mathbf{y}}_{\text{verb}}^{(2)}\}$ to the LLM, we serialize it into a JSON format.

**LLM for Verbalization:** The framework can be used with different LLMs. Let $LLM_{\text{verb}}$ represent the LLM responsible for generating verbalizations. The verbalization produced, denoted by $\mathbf{v}$, lies within the vocabulary space of $LLM_{\text{verb}}$.

**Prompt:** We assemble the serialized results into a prompt to the verbalizer $LLM_{\text{verb}}$. Our prompt is inspired by previous LLM work in XAI (Kroeger et al., 2023) and includes the following elements: *Context*, *Dataset*, *Task*, and *Instructions*, as illustrated in Box 1.

The *Context* outlines the type of models used, the classification task they perform, and a general overview of the dataset, including details about the features and the target variable. We choose to explicitly mention the feature names, $F = \{f_1, f_2, \ldots, f_d\}$, drawing insights from previous work (Hegselmann et al., 2023), which showed that feature names can help improve interpretability. We include the order of features in the representative sample to ensure that $LLM_{\text{verb}}$ can correctly associate feature names with their corresponding feature values. Additionally, we explicitly explain the meaning of the target variable, including what each possible value $c \in C$ represents.

The *Dataset* is the serialized representative sample, as described above.

The *Task* section states the underlying task we want $LLM_{\text{verb}}$ to perform.

The *Instructions* enumerate detailed instructions for the LLM.

---

**Context:** We have two logistic regression models trained on the same dataset for a binary classification task. The dataset contains details about random donors at a Blood Transfusion Service. The 4 features that it contains, in order, are: Recency (months), Frequency (times), Monetary (c.c. blood) and Time (months). The target feature (Blood Donated) is a binary variable representing whether the donor donated blood in March 2007 (1 stands for donating blood; 0 stands for not donating blood).

The dataset below contains a sample which includes the 4 input features in the order mentioned above as well as the outputs/predictions generated by each of the two models.

**Dataset:** ["input":[-66.287, -76.971, -76.971, -126.295], "output":{"model1":0, "model2": 0}, "input": [-66.287, 67.376, 67.376, -25.604],"output": {"model1": 1, "model2": 0} ... ]

**Task:** Based on the above sample set, generate a verbalization of the differences between the decision boundaries of the 2 models.

**Instructions:**

1. Go through the sample and analyze where the outputs differ and where they don't.

2. Identify the specific ranges of feature values for which the decision boundaries diverge. Provide these ranges in numerical terms, not just descriptive terms like 'high' or 'low'. Moreover, specify how the decisions of the two models diverge for these feature values.

3. Identify any features that appear to have a notable influence on the differences between the models' outputs.

4. Provide a clear and effective verbalization of how the decision boundaries of the two models diverge.

---

Box 1: Verbalization prompt template for LR models trained on the Blood dataset. It includes: *Context*, *Dataset*, *Task*, and *Instructions*.

## 4 EVALUATION

If a verbalization $\mathbf{v}$ accurately captures the differences between two models, it should facilitate an evaluator to predict the second model's outputs given the inputs and the outputs of the first.

We use an LLM to be the evaluator, and refer to it as $LLM_{\text{eval}}$. It uses the verbalization $\mathbf{v}$ to analyze an evaluation sample $\{\mathbf{X}_{\text{eval}}, \hat{\mathbf{y}}_{\text{eval}}^{(1)}\}$, which contains the input features $\mathbf{X}_{\text{eval}}$ and only the corresponding outputs of $M_1$, $\hat{\mathbf{y}}_{\text{eval}}^{(1)}$. $LLM_{\text{eval}}$ generates a simulated output $\tilde{\mathbf{y}}_{\text{eval}}^{(2)}$ corresponding to $\mathbf{X}_{\text{eval}}$. To assess the accuracy of simulated output, $\tilde{\mathbf{y}}_{\text{eval}}^{(2)}$, we use three evaluation metrics:

1. **Mismatch Accuracy** ($\mathbf{Acc}_{\text{mismatch}}$): It evaluates the cases where the outputs of $M_1$ and $M_2$ disagree, i.e., $I_{\text{mismatch}} = \{i \mid \hat{y}_{\text{eval},i}^{(1)} \neq \hat{y}_{\text{eval},i}^{(2)}\}$. For these cases, the accuracy is computed as proportion of cases where the simulated output matches that of $M_2$, i.e., $\tilde{y}_{\text{eval},i}^{(2)} = \hat{y}_{\text{eval},i}^{(2)}$, for $i \in I_{\text{mismatch}}$. The $\mathbf{Acc}_{\text{mismatch}}$ quantifies how well the verbalization $\mathbf{v}$ captures the points of divergence between the models.

2. **Match Accuracy** ($\mathbf{Acc}_{\text{match}}$): It considers the cases where the outputs of $M_1$ and $M_2$ agree, i.e., $I_{\text{match}} = \{i \mid \hat{y}_{\text{eval},i}^{(1)} = \hat{y}_{\text{eval},i}^{(2)}\}$. The accuracy is similarly computed as the proportion of these cases where the simulated output matches that of $M_2$. The $\mathbf{Acc}_{\text{match}}$ quantifies the extent of $\mathbf{v}$ introducing false differences between the models.

3. **Overall Accuracy** ($\mathbf{Acc}_{\text{overall}}$): This evaluates $\mathbf{v}$'s performance across all instances, combining both agreement and disagreement cases. It is computed as the overall proportion of cases where the synthetic output matches that of $M_2$.

The evaluation prompt template can be found in the appendix (see Appendix B).

## 5 EXPERIMENTAL SETUP

**Datasets:** We consider classification tasks, and based on prior work involving LLMs ((Hegselmann et al., 2023)), we selected the following three datasets: **Blood (784 rows, 4 features, 2 classes)**, **Diabetes (768 rows, 8 features, 2 classes)**, and **Car (1,728 rows, 6 features, 4 classes)**. The datasets were first divided into training and test sets. From the test set, we further split the data equally into two subsets: the $verb$ split, which is used as a representative sample for verbalization (as explained in Figure 3), and the $eval$ split, which is reserved for evaluation purposes. This ensures that verbalization and evaluation operate on distinct subsets.

To keep the input context manageable and ensure that each dataset had approximately 150 samples in both $verb$ and $eval$ splits, we adjusted the proportions of the initial train-test split. The train-test splits are shown in Table 1.

| Dataset | Train Split (%) | Test Split (%) |
|---------|-----------------|----------------|
| Blood | 60% | 40% |
| Diabetes | 60% | 40% |
| Car | 82% | 18% |

Table 1: Train-Test Split Percentages for Datasets

The datasets were scaled, and preprocessing steps were consistent across all model types.

**Models:** Through our experiments we study the performance of our framework across the two fundamental machine learning paradigms: model-based learning and instance-based learning. This complementary perspective spans different approaches to classification, while we anticipate poorer performance on instance-based algorithms due to their reliance on the entire training dataset and complex, data-dependent decision boundaries.

In the paradigm of model-based learning algorithms, we evaluate the efficacy of LLMs in verbalizing differences between two popular learning algorithms: (i) Logistic Regression (LR) and (ii) Decision Tree ( DT). We specifically chose these algorithms because they are widely used, interpretable and serve as good baselines in the development of LLM-based model management frameworks. To demonstrate the significant challenge of evaluating instance-based learning algorithms, we quantitatively demonstrate the difficulties faced by LLMs in verbalizing the difference for (iii) the K-Nearest Neighbors (KNNs) algorithm.

To streamline our study, we stratified the experiments based on the percentage of differing outputs between each pair of models, with three levels: (i) Level 1 ($15\% - 20\%$), (ii) Level 2 ($20\% - 25\%$), and (iii) Level 3 ($25\% - 30\%$). To measure the differences between models, we computed the percentage of differing outputs on the $verb$ split. For each of these levels, we generated multiple pairs of models for all three model types.

To generate pairs of LR models with a specific percentage of differing outputs, we first train a base model using `RandomizedSearchCV`. Then we create multiple variations by adding randomly generated noise to the base model's coefficients. The noise is controlled by a modification factor $m$ (noise $\sim \mathcal{N}(0, m\beta)$), where $\beta$ represents the vector of the base model's coefficients. We carefully adjust $m$ until the percentage of differing outputs between the base model and the modified model reaches the desired level. Rather than limiting our comparisons to the base model obtained from `RandomizedSearchCV`, we also compare the modified models against each other, identifying a diverse collection of model pairs.

We follow a similar process for Decision Trees and KNNs, with the details provided in Appendix A. For each model type and across all levels of output differences, we generate multiple base models and corresponding modified models.

**Verbalizers:** We include three state-of-the-art LLMs as $LLM_\text{verb}$: Claude 3.5 Sonnet (Anthropic, 2024), Gemini 1.5 Pro (Google, 2024), and GPT-4o (OpenAI, 2024). For each of these LLMs, we set the temperature as $T = 0.1$ in their respective API calls.

**Evaluator:** We let $LLM_\text{eval}$ be the same model as $LLM_\text{verb}$, to avoid the bias introduced when LLMs process the outputs of the other language models.

**Ablation Study on the effects of including model's internals:** The access to the internals, compared to solely relying on the representative samples, may help $LLM_{verb}$ understand (and therefore verbalize) how the models make decisions. We hypothesize that providing such model-specific information enables LLMs to generate more accurate and faithful verbalizations. We examine the effect of incorporating the models' internals on the performance of our framework in generating verbalizations. By internals, we refer to textual descriptions of a model's learned structure or information about its inner workings. Different model types have different key pieces of information that they rely upon to make predictions. For Logistic Regression, this entails providing the framework with the learned coefficients. For Decision Trees, we provide a textual representation of the learned structure, focusing on the decision rules and splits. Lastly, for completeness, we include KNNs, incorporating the number of neighbors (K) and the distance metric, as these parameters define their behavior.

**Ablation Study on the effects of excluding model-type:** The model-type is the name of the type of the model (e.g., Logistic Regression, Decision Tree, or KNN). We study the impact of excluding the model type when comparing models. We aim to evaluate if our framework can generate accurate verbalizations based purely on the observed behavior, rather than the names.

It should be noted that all our ablation studies are conducted using stratification Level 2 (20-25%) as the default configuration. Specific details about prompts can be found Appendix B.

| Model | Example Verbalization Excerpts |
|---|---|
| Claude | . . . The divergence is most pronounced when Recency is in the -90 to -65 range, Frequency and Monetary are very low (around -75 to -60), and Time is very low (-135 to -80). In these scenarios, Model 1 predicts a positive outcome, while Model 2 predicts a negative outcome . . . |
| GPT | . . . The decision boundaries of the two logistic regression models diverge primarily in the negative ranges of the features Recency, Frequency, Monetary, and Time. Specifically, when the values of these features are within the ranges of approximately -90 to -60 for Recency, -77 to -61 for Frequency and Monetary, and -135 to -126 for Time, Model 2 is more likely to predict a positive outcome (1) compared to Model 1, which tends to predict a negative outcome (0) . . . |
| Gemini | . . . Model 2 appears to predict '1' (donating blood) more often than Model 1 for certain ranges of feature values. Specifically, Model 2 predicts '1' when:

- **Frequency** is positive and **Time** is between approximately -25 and 40 months, and **Recency** is less than approximately -70 months . . . |

Table 2: Verbalization Excerpts for a pair of Level 2 (20%-25%) LR models trained on the Blood Dataset.

| Model | Example Verbalization Excerpts |
|---|---|
| Claude | . . . Model 2 tends to be more lenient in its classifications compared to Model 1, often assigning higher class values.

The most significant divergence occurs when the buying price (first feature) is 0 or 1, maintenance price (second feature) is 0 or 1, and safety (sixth feature) is 2. In these cases, Model 2 often assigns a class of 2, while Model 1 assigns a class of 0 or 1 . . . |
| GPT | . . . When the buying price is low (0), Model 2 tends to predict higher classes (more favorable evaluations) compared to Model 1, especially when other conditions such as safety, capacity, and maintenance are moderate to high.

Similar to buying price, lower maintenance costs (0) often lead to higher predictions in Model 2 compared to Model 1, under favorable conditions in other features . . . |
| Gemini | . . . **Model 2 demonstrates greater leniency towards cars with higher maintenance costs.** Specifically, when the maintenance price is "high" (encoded as 2) or "very high" (encoded as 3), Model 2 tends to assign a higher evaluation class compared to Model 1, even when other features like buying price are less favorable . . . |

Table 3: Verbalization Excerpts for a pair of Level 2 (20%-25%) DT models trained on the Car Dataset.

# 6 RESULTS

## 6.1 COMPARING LOGISTIC REGRESSORS

Our framework demonstrates strong performance when applied to logistic regression across datasets, likely due to their linear nature. Figure 2a shows the performance on LR models trained on the Blood and Car datasets. Among the 3 LLMs, Claude 3.5 Sonnet achieves the best performance, with a $\mathbf{Acc}_{\text{mismatch}}$ of $0.831 \pm 0.016$ and a $\mathbf{Acc}_{\text{match}}$ of $0.860 \pm 0.018$, indicating its ability to effectively articulate the points of divergence without introducing any false differences. GPT-4o follows closely with slightly lower yet competitive results, achieving a $\mathbf{Acc}_{\text{mismatch}}$ of $0.779 \pm 0.026$ and a $\mathbf{Acc}_{\text{match}}$ of $0.822 \pm 0.020$. Gemini lags behind, obtaining a $\mathbf{Acc}_{\text{mismatch}}$ of $0.676 \pm 0.027$ and a $\mathbf{Acc}_{\text{match}}$ of $0.820 \pm 0.023$. This indicates significant variation in how well each LLM handles the task of verbalization for a pair of logistic regression models.

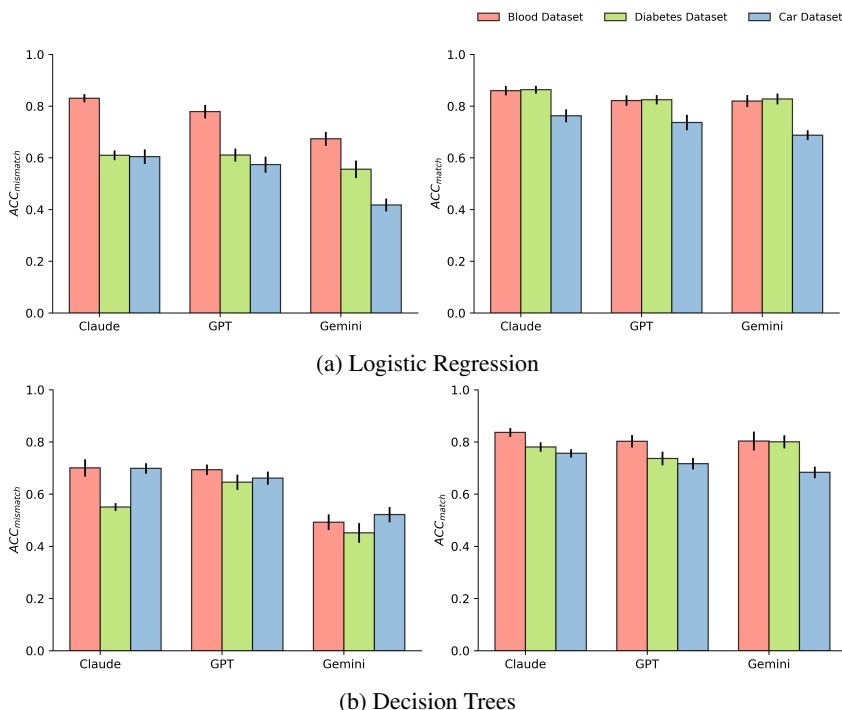

(a) Logistic Regression

(b) Decision Trees

Figure 2: Performance of three LLMs. 2a shows the $\mathbf{Acc}_{\text{mismatch}}$ and $\mathbf{Acc}_{\text{match}}$ for Level 2 $(20\% - 25\%)$ LR models trained on Blood, Diabetes, and Car datasets. 2b shows the same for DTs.

Performance decreases across all datasets at the most challenging level, Level 1 (15-20%), as detailed in Table 4. This suggests that as the problem complexity increases, even the best-performing LLMs can't keep up the same level of accuracy.

For the Diabetes and Car dataset, we observe a drop in the performance of the framework, which can be attributed to the increasing complexity of the datasets - Diabetes with a larger number of features and Car with multiple classes. Nevertheless, both Claude and GPT-4o maintain $\mathbf{Acc}_{\text{mismatch}}$ of $0.605{\pm}0.028$ and $0.574{\pm}0.031$ respectively for the Car dataset, indicating that their performance remains substantially above the random-guessing baseline. These results suggest that LLMs are effective at verbalizing differences between logistic regression models. Table 2 shows excerpts from some of these verbalizations.

## 6.2 COMPARING DECISION TREES

Decision Trees present a difficult challenge compared to LR models, mainly due to their non-linear decision boundaries. Consequently, the framework's performance when applied to DTs is lower, although similar trends from LR are observed.

Figure 2b illustrates that, on the Blood dataset, Claude 3.5 Sonnet remains the top performer, with a $\mathbf{Acc}_{\text{mismatch}}$ of $0.700{\pm}0.03$ and $\mathbf{Acc}_{\text{match}}$ of $0.837{\pm}0.017$. While competitive, GPT-4o's results are slightly lower than Claude's, with a $\mathbf{Acc}_{\text{mismatch}}$ of $0.694{\pm}0.020$ and $\mathbf{Acc}_{\text{match}}$ of $0.803{\pm}0.024$. In contrast, Gemini performs notably worse, with a particularly low $\mathbf{Acc}_{\text{mismatch}}$ of $0.493{\pm}0.030$, highlighting its difficulties in capturing points of divergence.

The Car dataset introduces additional complexity. Claude's performance drops slightly but remains strong, with $\mathbf{Acc}_{\text{mismatch}}$ of $0.700{\pm}0.020$ and $\mathbf{Acc}_{\text{match}}$ of $0.757{\pm}0.016$. GPT-4o displays a similar decline in its performance with $\mathbf{Acc}_{\text{mismatch}}$ of $0.662{\pm}0.025$ and $\mathbf{Acc}_{\text{match}}$ of $0.717{\pm}0.022$. Gemini's results are again the lowest, with indicating its difficulty in distinguishing between DTs.

Despite the drop in overall performance for DTs across the datasets, Claude and GPT-4o manage to maintain a relatively strong performance. These findings suggest a broader trend: LLMs are generally

able to verbalize the difference between DTs effectively. Table 3 shows excerpts from some of these verbalizations.

## 6.3 COMPARING KNNS

KNNs appear more challenging for our framework due to their instance-based learning nature. Given their reliance on specific local data points for predictions, we anticipate that our Model Manager struggles to effectively verbalize instance-based learning algorithms, and our observations support the anticipation. For Level 2 $(20\% - 25\%)$ models on the Blood dataset, the $\mathbf{Acc}_{\text{mismatch}}$ scores were lower than $0.7$, with Gemini lower than $0.6$. On the Car and Diabetes datasets, the performance declines further, with Claude and GPT-4o failing to surpass $0.50$ for $\mathbf{Acc}_{\text{mismatch}}$. We include the complete details of the KNN experiments in Table 6.

## 6.4 ABLATION STUDIES

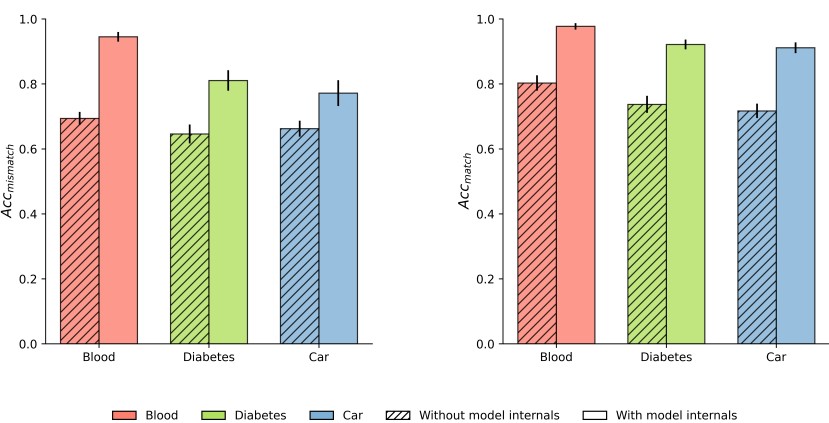

Figure 3: Comparison of GPT-4o's performance on DTs, with and without models' internals, for the Blood, Diabetes, and Car datasets. Including models' internals resulted in performance improvements across all cases.

**a) Effects of including models' internals:** For LR, the inclusion of coefficients results in either performance remaining within the error margin or showing a modest increase (3-5%) across all datasets. This suggests that while the coefficients may help LLMs to better understand feature importances, the relatively simple nature of logistic regression means the gains are minimal.

The most pronounced impact of including models' internals can be seen for Decision Trees (Figure 3). For the Blood Dataset, GPT-4o's performance jumps to a $\mathbf{Acc}_{\text{mismatch}}$ of $0.945\pm0.015$ and $\mathbf{Acc}_{\text{match}}$ of $0.971\pm0.01$, representing a 23.81% increase in $\mathbf{Acc}_{\text{overall}}$. For Claude it increases to a $\mathbf{Acc}_{\text{mismatch}}$ of $0.747\pm0.029$ and $\mathbf{Acc}_{\text{match}}$ of $0.879\pm0.018$. Even Gemini shows a notable increase, reaching to an $\mathbf{Acc}_{\text{mismatch}}$ of $0.747\pm0.03$ and an $\mathbf{Acc}_{\text{match}}$ of $0.852\pm0.026$. Similar trends were observed across the other datasets, with GPT-4o showing a 25.4% improvement on the Diabetes dataset, while the Car dataset exhibited more moderate but still meaningful gains.

These findings indicate that decision trees' rule-based nature likely enables LLMs to better capture and articulate the model's underlying decision-making process. The explicit structure of decision paths in decision trees seems to facilitate more accurate and interpretable verbalizations.

As hypothesized, KNN models showed minimal or even slightly negative effects when model-specific information was included. This reinforces the idea that KNN's reliance on local instance-based learning, rather than explicit parameters or decision rules, poses challenges for LLMs in verbalizing model behavior effectively. The slight negative effect can be attributed to LLM focusing on the parameters passed and not the sample set.

The impact of including model-specific information varied depending upon the type of model. For logistic regression, a marginal increase was observed in the scores. However, decision trees witness the most substantial improvement, with performance gains across all datasets and all LLMs, with $\text{Acc}_{\text{overall}}$ even reaching above 0.9 in some cases. This underscores the effectiveness of including model-specific information in generating more accurate and faithful verbalizations. These findings suggest a broader trend: For certain model types, including model-specific information can significantly enhance the quality of generated verbalizations.

**b) Effects of excluding model-type:**   The results in subsection A.2 show that removing model-type information from the prompt had little effect on the quality of verbalizations, with performance variations remaining within the margin of error. This implies that our framework relies mainly on the observed behavior (i.e., the representative sample) when verbalizing differences in decision boundaries.

## 7 DISCUSSION

Our results show promising trends when verbalizing the differences of parametric models (LR and DT). The non-parametric KNN models, on the other hand, introduce more challenges, as indicated by the lower $\text{Acc}_{\text{match}}$ and $\text{Acc}_{\text{mismatch}}$. On one hand, these indicate that future Model Managers on non-parametric models need to consider factors that describe the dataset. On the other hand, these indicate that the Model Managers can be extended to verbalizing the differences between Deep Neural Networks, especially incorporating approaches that describe the models' internals (e.g., mechanistic interpretability). Considering the complex nature of DNNs, the developers for Model Managers on DNNs will have to consider a lot of intricate details.

The plug-and-compare flexibility of Model Manager allows potential upgrades to the Manager. When newer, higher-capability LMs are developed, we can replace the LM in Model Manager with the next-generation ones. The same flexibility applies to the prompting techniques and the expected tasks (for example, comparing across more than two models).

A good resource manager does not just observe. Beyond verbalization, a fully-fledged Model Manager should be able to automatically inspect the individual models, question the potential weaknesses, and potentially recommend improvement methods, including but not limited to model merging, model safeguarding, and model debiasing. A lot of future work is needed toward this goal, which we believe deserves more attention from the field.

## 8 CONCLUSION

In conclusion, the Model Manager framework establishes a foundational step toward automatic management of machine learning models. The Model Manager verbalizes the difference between two models. While it excels in identifying differences between parametric models, challenges remain with non-parametric models like KNNs, highlighting the need for tailored strategies that accommodate the unique characteristics of various model types. This research sets the stage for future research in model management tools that can dynamically adapt to the evolving landscape of ML technologies. As we look to the future, integrating more sophisticated language models and expanding the framework's capabilities will be essential in advancing the field towards more transparent, accountable, and effective AI systems.

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

## A    APPENDIX: EXPERIMENT DETAILS AND FULL RESULTS

### A.1    ADDITIONAL EXPERIMENTAL DETAILS

**DT generation:** For DTs, similar to the LR models, we first train a base model using `RandomizedSearchCV`. To generate a modified DT, we introduce two levels of variation. First, we randomly sample new hyperparameters from the defined space. This ensures that the modified tree has a structure different from the base model. Second, we add noise to the splitting thresholds of the nodes. The noise is normally distributed and controlled by a modification factor $m$ (noise $\sim \mathcal{N}(0, m)$) and is scaled to the level of thresholds (noise $= \tau * $ noise ). We carefully adjust $m$ until the percentage of differing outputs between the base model and the modified model reaches the desired level. Rather than limiting our comparison to the base model obtained from `RandomizedSearchCV`, we also compare the modified models against each other, identifying a diverse collection of pairs.

**KNNs generation**    : In the case of KNNs, we first train a base model using `RandomizedSearchCV`. To generate modified versions, we randomly sample new hyperparameters and compare the predictions of the base model with each modified model, calculating the percentage of differing outputs until it reaches the desired level. Additionally, we compare the modified models against each other to obtain a diverse collection of pairs.

### A.2    FULL EXPERIMENTAL RESULTS

We present complete results for these models in Table 4, Table 5 and Table 6.

Table 4: Evaluation metrics for LR models across different datasets. Each row includes the performance metrics for an LLM, measured across Level 1 (15%-20%), Level 2 (20%-25%), Level 3 $(25\% - 30\%)$, Level 4 $(20\% - 25\%$ With Models' Internals), and Level 5 $(20\% - 25\%$ Without Model Type).

| LLM | Metric | Level 1 | Level 2 | Level 3 | Level 4 | Level 5 |
|---|---|---|---|---|---|---|
| **Blood Dataset** | | | | | | |
| Claude 3.5 Sonnet | $\mathbf{Acc}_{\text{mismatch}}$ | 0.806 ±.021 | 0.831 ±.016 | 0.871 ±.009 | 0.869 ±.019 | 0.828 ±.014 |
| | $\mathbf{Acc}_{\text{match}}$ | 0.697 ±.012 | 0.860 ±.018 | 0.808 ±.015 | 0.844 ±.014 | 0.861 ±.023 |
| | $\mathbf{Acc}_{\text{overall}}$ | 0.717 ±.009 | 0.854 ±.016 | 0.824 ±.09 | 0.850 ±.013 | 0.854 ±.019 |
| GPT-4o | $\mathbf{Acc}_{\text{mismatch}}$ | 0.744 ±.016 | 0.779 ±.026 | 0.763 ±.013 | 0.804 ±.020 | 0.780 ±.025 |
| | $\mathbf{Acc}_{\text{match}}$ | 0.804 ±.016 | 0.822 ±.020 | 0.828 ±.013 | 0.812 ±.015 | 0.839 ±.018 |
| | $\mathbf{Acc}_{\text{overall}}$ | 0.794 ±.013 | 0.815 ±.015 | 0.809 ±.009 | 0.811 ±.013 | 0.827 ±.014 |
| Gemini 1.5 Pro | $\mathbf{Acc}_{\text{mismatch}}$ | 0.670 ±.033 | 0.674 ±.027 | 0.710 ±.021 | 0.663 ±.030 | 0.716 ±.023 |
| | $\mathbf{Acc}_{\text{match}}$ | 0.761 ±.022 | 0.820 ±.023 | 0.760 ±.020 | 0.854 ±.024 | 0.793 ±.029 |
| | $\mathbf{Acc}_{\text{overall}}$ | 0.747 ±.016 | 0.776 ±.019 | 0.744 ±.013 | 0.816 ±.021 | 0.774 ±.023 |
| **Car Dataset** | | | | | | |
| Claude 3.5 Sonnet | $\mathbf{Acc}_{\text{mismatch}}$ | 0.612 ±.025 | 0.605 ±.028 | 0.711 ±.021 | 0.655 ±.020 | 0.602 ±.033 |
| | $\mathbf{Acc}_{\text{match}}$ | 0.741 ±.022 | 0.763 ±.025 | 0.802 ±.026 | 0.762 ±.021 | 0.758 ±.034 |
| | $\mathbf{Acc}_{\text{overall}}$ | 0.718 ±.017 | 0.725 ±.018 | 0.776 ±.020 | 0.735 ±.016 | 0.719 ±.024 |
| GPT-4o | $\mathbf{Acc}_{\text{mismatch}}$ | 0.541 ±.026 | 0.574 ±.031 | 0.608 ±.024 | 0.629 ±.020 | 0.557 ±.031 |
| | $\mathbf{Acc}_{\text{match}}$ | 0.713 ±.027 | 0.737 ±.030 | 0.771 ±.023 | 0.762 ±.020 | 0.745 ±.033 |
| | $\mathbf{Acc}_{\text{overall}}$ | 0.679 ±.023 | 0.697 ±.022 | 0.729 ±.015 | 0.729 ±.016 | 0.699 ±.023 |
| Gemini 1.5 Pro | $\mathbf{Acc}_{\text{mismatch}}$ | 0.416 ±.014 | 0.418 ±.025 | 0.446 ±.016 | 0.417 ±.023 | 0.406 ±.021 |
| | $\mathbf{Acc}_{\text{match}}$ | 0.693 ±.024 | 0.688 ±.019 | 0.606 ±.032 | 0.755 ±.017 | 0.690 ±.022 |
| | $\mathbf{Acc}_{\text{overall}}$ | 0.638 ±.018 | 0.624 ±.016 | 0.562 ±.023 | 0.674 ±.014 | 0.624 ±.019 |
| **Diabetes Dataset** | | | | | | |
| Claude 3.5 Sonnet | $\mathbf{Acc}_{\text{mismatch}}$ | 0.522 ±.040 | 0.610 ±.019 | 0.616 ±.025 | 0.619 ±.017 | 0.600 ±.026 |
| | $\mathbf{Acc}_{\text{match}}$ | 0.777 ±.024 | 0.864 ±.015 | 0.831 ±.021 | 0.874 ±.012 | 0.884 ±.017 |
| | $\mathbf{Acc}_{\text{overall}}$ | 0.702 ±.017 | 0.805 ±.011 | 0.772 ±.018 | 0.815 ±.008 | 0.820 ±.013 |
| GPT-4o | $\mathbf{Acc}_{\text{mismatch}}$ | 0.442 ±.030 | 0.611 ±.025 | 0.544 ±.027 | 0.628 ±.022 | 0.617 ±.021 |
| | $\mathbf{Acc}_{\text{match}}$ | 0.687 ±.023 | 0.825 ±.018 | 0.687 ±.025 | 0.829 ±.011 | 0.846 ±.018 |
| | $\mathbf{Acc}_{\text{overall}}$ | 0.642 ±.016 | 0.776 ±.015 | 0.645 ±.020 | 0.786 ±.008 | 0.791 ±.013 |
| Gemini 1.5 Pro | $\mathbf{Acc}_{\text{mismatch}}$ | 0.398 ±.023 | 0.556 ±.034 | 0.454 ±.029 | 0.583 ±.034 | 0.564 ±.026 |
| | $\mathbf{Acc}_{\text{match}}$ | 0.808 ±.016 | 0.828 ±.021 | 0.671 ±.032 | 0.855 ±.021 | 0.814 ±.025 |
| | $\mathbf{Acc}_{\text{overall}}$ | 0.723 ±.013 | 0.768 ±.015 | 0.607 ±.024 | 0.800 ±.015 | 0.756 ±.020 |

Table 5: Evaluation metrics for DT models across different datasets. Each row includes the performance metrics for an LLM, measured across Level 1 (15%-20%), Level 2 (20%-25%), Level 3 $(25\% - 30\%)$, Level 4 $(20\% - 25\%$ With Models' Internals), and Level 5 $(20\% - 25\%$ Without Model Type).

| LLM | Metric | Level 1 | Level 2 | Level 3 | Level 4 | Level 5 |
|---|---|---|---|---|---|---|
| **Blood Dataset** | | | | | | |
| Claude 3.5 Sonnet | $\text{Acc}_{\text{mismatch}}$ | 0.654 ±.015 | 0.701 ±.033 | 0.788 ±.024 | 0.747 ±.029 | 0.699 ±.029 |
| | $\text{Acc}_{\text{match}}$ | 0.861 ±.019 | 0.837 ±.017 | 0.861 ±.023 | 0.879 ±.018 | 0.849 ±.018 |
| | $\text{Acc}_{\text{overall}}$ | 0.826 ±.018 | 0.812 ±.010 | 0.834 ±.015 | 0.854 ±.017 | 0.822 ±.017 |
| GPT-4o | $\text{Acc}_{\text{mismatch}}$ | 0.693 ±.029 | 0.694 ±.020 | 0.758 ±.022 | 0.945 ±.015 | 0.699 ±.023 |
| | $\text{Acc}_{\text{match}}$ | 0.823 ±.025 | 0.803 ±.024 | 0.838 ±.022 | 0.971 ±.010 | 0.805 ±.019 |
| | $\text{Acc}_{\text{overall}}$ | 0.800 ±.022 | 0.780 ±.019 | 0.808 ±.015 | 0.966 ±.009 | 0.783 ±.017 |
| Gemini 1.5 Pro | $\text{Acc}_{\text{mismatch}}$ | 0.521 ±.021 | 0.493 ±.030 | 0.739 ±.041 | 0.747 ±.030 | 0.499 ±.025 |
| | $\text{Acc}_{\text{match}}$ | 0.817 ±.029 | 0.804 ±.036 | 0.852 ±.020 | 0.852 ±.026 | 0.793 ±.022 |
| | $\text{Acc}_{\text{overall}}$ | 0.764 ±.024 | 0.737 ±.027 | 0.818 ±.017 | 0.832 ±.020 | 0.729 ±.021 |
| **Car Dataset** | | | | | | |
| Claude 3.5 Sonnet | $\text{Acc}_{\text{mismatch}}$ | 0.599 ±.028 | 0.699 ±.020 | 0.680 ±.026 | 0.732 ±.039 | 0.700 ±.024 |
| | $\text{Acc}_{\text{match}}$ | 0.753 ±.022 | 0.757 ±.016 | 0.772 ±.020 | 0.823 ±.024 | 0.753 ±.017 |
| | $\text{Acc}_{\text{overall}}$ | 0.721 ±.013 | 0.743 ±.014 | 0.748 ±.015 | 0.802 ±.024 | 0.740 ±.015 |
| GPT-4o | $\text{Acc}_{\text{mismatch}}$ | 0.599 ±.025 | 0.662 ±.025 | 0.620 ±.028 | 0.772 ±.040 | 0.663 ±.024 |
| | $\text{Acc}_{\text{match}}$ | 0.778 ±.018 | 0.717 ±.022 | 0.794 ±.019 | 0.911 ±.016 | 0.720 ±.019 |
| | $\text{Acc}_{\text{overall}}$ | 0.745 ±.014 | 0.703 ±.019 | 0.749 ±.017 | 0.882 ±.014 | 0.706 ±.015 |
| Gemini 1.5 Pro | $\text{Acc}_{\text{mismatch}}$ | 0.483 ±.028 | 0.522 ±.029 | 0.510 ±.000 | 0.567 ±.037 | 0.528 ±.034 |
| | $\text{Acc}_{\text{match}}$ | 0.721 ±.026 | 0.684 ±.022 | 0.699 ±.000 | 0.835 ±.013 | 0.678 ±.028 |
| | $\text{Acc}_{\text{overall}}$ | 0.677 ±.021 | 0.651 ±.020 | 0.652 ±.000 | 0.774 ±.016 | 0.647 ±.023 |
| **Diabetes Dataset** | | | | | | |
| Claude 3.5 Sonnet | $\text{Acc}_{\text{mismatch}}$ | 0.479 ±.019 | 0.551 ±.015 | 0.610 ±.019 | 0.657 ±.033 | 0.553 ±.021 |
| | $\text{Acc}_{\text{match}}$ | 0.828 ±.018 | 0.781 ±.018 | 0.843 ±.021 | 0.913 ±.014 | 0.773 ±.022 |
| | $\text{Acc}_{\text{overall}}$ | 0.752 ±.016 | 0.736 ±.016 | 0.785 ±.013 | 0.864 ±.014 | 0.730 ±.018 |
| GPT-4o | $\text{Acc}_{\text{mismatch}}$ | 0.548 ±.017 | 0.646 ±.029 | 0.566 ±.031 | 0.811 ±.032 | 0.652 ±.026 |
| | $\text{Acc}_{\text{match}}$ | 0.786 ±.019 | 0.737 ±.026 | 0.815 ±.020 | 0.921 ±.015 | 0.747 ±.022 |
| | $\text{Acc}_{\text{overall}}$ | 0.734 ±.014 | 0.719 ±.019 | 0.754 ±.015 | 0.902 ±.015 | 0.728 ±.020 |
| Gemini 1.5 Pro | $\text{Acc}_{\text{mismatch}}$ | 0.441 ±.031 | 0.452 ±.038 | 0.611 ±.040 | 0.590 ±.357 | 0.528 ±.34 |
| | $\text{Acc}_{\text{match}}$ | 0.822 ±.024 | 0.801 ±.025 | 0.899 ±.014 | 0.851 ±.236 | 0.678 ±.28 |
| | $\text{Acc}_{\text{overall}}$ | 0.739 ±.019 | 0.719 ±.016 | 0.826 ±.013 | 0.801 ±.217 | 0.647 ±.23 |

Table 6: Evaluation metrics for KNN models across different datasets. Each row includes the performance metrics for an LLM, measured across Level 1 (15%-20%), Level 2 (20%-25%), Level 3 ($25\% - 30\%$), Level 4 ($20\% - 25\%$ With Models' Internals), and Level 5 ($20\% - 25\%$ Without Model Type).

| LLM | Metric | Level 1 | Level 2 | Level 3 | Level 4 | Level 5 |
|---|---|---|---|---|---|---|
| **Blood Dataset** | | | | | | |
| Claude 3.5 Sonnet | $\text{Acc}_{\text{mismatch}}$ | 0.656 ±.021 | 0.686 ±.023 | 0.777 ±.031 | 0.720 ±.024 | 0.707 ±.022 |
| | $\text{Acc}_{\text{match}}$ | 0.826 ±.020 | 0.845 ±.024 | 0.717 ±.020 | 0.847 ±.023 | 0.832 ±.028 |
| | $\text{Acc}_{\text{overall}}$ | 0.795 ±.019 | 0.811 ±.019 | 0.737 ±.011 | 0.821 ±.018 | 0.805 ±.022 |
| GPT-4o | $\text{Acc}_{\text{mismatch}}$ | 0.647 ±.019 | 0.648 ±.023 | 0.722 ±.019 | 0.708 ±.019 | 0.663 ±.029 |
| | $\text{Acc}_{\text{match}}$ | 0.856 ±.019 | 0.876 ±.015 | 0.776 ±.020 | 0.836 ±.031 | 0.873 ±.018 |
| | $\text{Acc}_{\text{overall}}$ | 0.818 ±.017 | 0.829 ±.014 | 0.767 ±.015 | 0.809 ±.023 | 0.830 ±.015 |
| Gemini 1.5 Pro | $\text{Acc}_{\text{mismatch}}$ | 0.549 ±.030 | 0.559 ±.031 | 0.608 ±.025 | 0.576 ±.036 | 0.564 ±.031 |
| | $\text{Acc}_{\text{match}}$ | 0.687 ±.023 | 0.774 ±.020 | 0.709 ±.024 | 0.802 ±.025 | 0.757 ±.020 |
| | $\text{Acc}_{\text{overall}}$ | 0.662 ±.022 | 0.729 ±.019 | 0.686 ±.021 | 0.755 ±.020 | 0.717 ±.019 |
| **Car Dataset** | | | | | | |
| Claude 3.5 Sonnet | $\text{Acc}_{\text{mismatch}}$ | 0.454 ±.016 | 0.490 ±.030 | 0.499 ±.014 | 0.469 ±.030 | 0.477 ±.031 |
| | $\text{Acc}_{\text{match}}$ | 0.760 ±.017 | 0.709 ±.032 | 0.752 ±.025 | 0.616 ±.046 | 0.654 ±.033 |
| | $\text{Acc}_{\text{overall}}$ | 0.705 ±.013 | 0.657 ±.029 | 0.688 ±.019 | 0.581 ±.040 | 0.613 ±.030 |
| GPT-4o | $\text{Acc}_{\text{mismatch}}$ | 0.345 ±.024 | 0.460 ±.031 | 0.455 ±.023 | 0.411 ±.026 | 0.466 ±.039 |
| | $\text{Acc}_{\text{match}}$ | 0.828 ±.012 | 0.751 ±.030 | 0.773 ±.020 | 0.651 ±.039 | 0.724 ±.025 |
| | $\text{Acc}_{\text{overall}}$ | 0.737 ±.010 | 0.682 ±.029 | 0.692 ±.015 | 0.593 ±.033 | 0.665 ±.025 |
| Gemini 1.5 Pro | $\text{Acc}_{\text{mismatch}}$ | 0.304 ±.021 | 0.325 ±.026 | 0.353 ±.019 | 0.332 ±.034 | 0.330 ±.025 |
| | $\text{Acc}_{\text{match}}$ | 0.593 ±.029 | 0.626 ±.034 | 0.672 ±.024 | 0.629 ±.026 | 0.625 ±.023 |
| | $\text{Acc}_{\text{overall}}$ | 0.536 ±.024 | 0.554 ±.030 | 0.591 ±.019 | 0.558 ±.023 | 0.554 ±.021 |
| **Diabetes Dataset** | | | | | | |
| Claude 3.5 Sonnet | $\text{Acc}_{\text{mismatch}}$ | 0.616 ±.014 | 0.603 ±.025 | 0.624 ±.013 | 0.589 ±.024 | 0.606 ±.033 |
| | $\text{Acc}_{\text{match}}$ | 0.840 ±.020 | 0.800 ±.029 | 0.716 ±.025 | 0.758 ±.036 | 0.805 ±.030 |
| | $\text{Acc}_{\text{overall}}$ | 0.796 ±.017 | 0.756 ±.024 | 0.693 ±.022 | 0.720 ±.030 | 0.762 ±.028 |
| GPT-4o | $\text{Acc}_{\text{mismatch}}$ | 0.626 ±.030 | 0.566 ±.027 | 0.556 ±.019 | 0.519 ±.022 | 0.490 ±.031 |
| | $\text{Acc}_{\text{match}}$ | 0.864 ±.019 | 0.784 ±.032 | 0.702 ±.041 | 0.792 ±.020 | 0.763 ±.032 |
| | $\text{Acc}_{\text{overall}}$ | 0.819 ±.018 | 0.736 ±.026 | 0.664 ±.031 | 0.733 ±.017 | 0.705 ±.029 |
| Gemini 1.5 Pro | $\text{Acc}_{\text{mismatch}}$ | 0.422 ±.022 | 0.473 ±.029 | 0.510 ±.029 | 0.462 ±.032 | 0.460 ±.034 |
| | $\text{Acc}_{\text{match}}$ | 0.852 ±.017 | 0.774 ±.030 | 0.699 ±.031 | 0.782 ±.028 | 0.767 ±.027 |
| | $\text{Acc}_{\text{overall}}$ | 0.770 ±.015 | 0.709 ±.026 | 0.650 ±.024 | 0.713 ±.021 | 0.701 ±.023 |

# B PROMPTS

---

**Context:** We have two {MODEL_TYPE} models trained on the same dataset for a {CLASSIFICATION_TYPE} problem. {DATASET_DESCRIPTION}

The verbalization below contains a natural language description of the differences between the decision boundaries of the two models.

**Dataset:** {DATASET_SAMPLE}

**Verbalization:** {VERBALIZATION}

**Task:** Based on the above verbalization, predict the output of Model 2 for each of the input instance in the above sample.

**Instructions:** Think about the question carefully. Go through the verbalization thoroughly. Analyze the input features in the sample. After explaining your reasoning, provide the answer in a JSON format within markdown at the end. The JSON should contain the input features and the output of Model 2. Do not provide any further details after the JSON.

---

Box 2: Evaluation Prompt Template

---

**Context**: We have two {MODEL_TYPE} models trained on the same dataset for a {CLASSIFICATION_PROBLEM} task. {DATSET_DESCRIPTION}

**Model Information**: {MODEL_INFO}

**Dataset**: {DATASET_SAMPLE}

**Task**: Based on the above model information and the sample set, generate a verbalization of the differences between the decision boundaries of the two models.

**Instructions**:

1. Review the model information and go through the sample. Analyze where the outputs differ and where they don't.

2. Identify the specific ranges of feature values for which the decision boundaries diverge. Provide these ranges in numerical terms, not just descriptive terms like 'high' or 'low'. Moreover, specify how the decisions of the two models diverge for these feature values.

3. Identify any features that appear to have a notable influence on the differences between the models' outputs.

4. Provide a clear and effective verbalization of how the decision boundaries of the two models diverge.

---

Box 3: Ablation Study 1 Prompt Template (Effects of Including Model Information)

**Context**: We have two models trained on the same dataset for a {CLASSIFICAITON_PROBLEM} task. {DATASET_DESCRIPTION}

**Dataset**: {DATASET_SAMPLE}

**Task**: Based on the above set, generate a verbalization of the differences between the decision boundaries of the two models.

**Instructions**:

1. Go through the sample and analyze where the outputs differ and where they don't.

2. Identify the specific ranges of feature values for which the decision boundaries diverge. Provide these ranges in numerical terms, not just descriptive terms like 'high' or 'low'. Moreover, specify how the decisions of the two models diverge for these feature values.

3. Identify any features that appear to have a notable influence on the differences between the models' outputs.

4. Provide a clear and effective verbalization of how the decision boundaries of the two models diverge.

Box 4: Ablation Study 2 Prompt Template (Effects of Removing Model Type)

