# OpenReview forum: "A Language Model based Model Manager"
_ICLR.cc/2025/Conference — Submitted to ICLR 2025_

### Official Review · Reviewer_KAiZ · 2024-10-27

**Soundness:** 2
**Presentation:** 2
**Contribution:** 2
**Rating:** 5
**Confidence:** 4

**Summary:**

The paper focuses on helping practitioners distinguish between models trained to accomplish the same task. Specifically, the authors design a method to produce natural language explanations of differences between models using model predictions on a set of examples. The method is validated in the context of pairs of logistic regressions, decision trees, and KNNs, and they measure the quality of the natural language explanation by measuring how an LLM can reconstruct one model’s predictions given only the difference explanation and the other model’s predictions. Experiments show that the quality of the explanations depends on model type, model description, and the extent to which the models differ.

**Strengths:**

The authors tackle a common yet understudied problem and develop a unique solution. The paper was generally clear and easy to follow. With expanded experimentation and detail around methodology, I believe it could make a strong contribution.

**Weaknesses:**

I see a few high-level limitations (L1-L3) and minor limitations (L4-L6) with the experimentation.

L1. The presented experiments are limited to models that are not typically associated with the “model lake” phenomenon. I’m also not sure that results describing model differences using logistic regressions and decision trees immediately generalize to deep learning models, since the space of verbalizations is likely much more complex with the latter.

L2. The prompt implies there exists some verbalization that explains differences in model predictions. But sometimes, model disagreements could be due to random chance, rather than some systematic, explainable difference. The same could happen if the verbalization dataset is too small. I’d be interested to see the verbalizations for two models which exhibit the same behavior, but produce slightly different predictions because of some other source of randomness (e,g. being trained on two distinct datasets). Is this a case the proposed approach handles? What do verbalizations look like in this setting?

L3. The absence of baselines made it difficult to contextualize results. You could replace LLM_{verb} with an interpretable-by-design model; for example, what if you trained a logistic regression to predict Model 2’s predictions on an example given the example’s features and Model 1’s predictions? How does that do in comparison to reconstructing Model 2’s predictions given the verbalization using LLM_{verb}?

L4. The reliance on feature names makes it difficult to generalize the methodology to settings in which models make use of high-dimensional inputs with no natural feature names (e.g. images).

L5. Changing both LLM_{verb} and LLM_{eval} at the same time makes it difficult to assess whether results are different because a certain LLM is a better verbalizer or a better evaluator. I’m not as familiar with the biases of passing one LLM’s outputs as input to another, but it may still be useful to hold LLM_{eval} constant to isolate the performance of each LLM as a verbalizer.

L6. It’s worth clarifying early on that the examples used to produce a verbalization are distinct from the examples used to train each model. I was a bit confused by “It does so by serializing a representative sample of input instances (from the dataset) and the corresponding model outputs in a JSON format.“ in the introduction, but later clarified my confusion when reading the problem setting. I’d move this information so that it appears earlier.

There are a few citations you might consider including; the following seem relevant:

1. https://arxiv.org/abs/2201.12323 – a method to describe differences in text distributions. Seems like such a method could be used to describe differences between LLMs applied to a given task.
2. https://arxiv.org/abs/2110.10545 - early work on choosing the “best” model from a pre-trained model hub
3. https://arxiv.org/pdf/2404.04326 - the authors here use an LLM to generate hypotheses; one could frame the task LLM_{verb} attempts to solve as hypothesis generation, where the model is developing a natural language hypothesis to describe Model 2’s predictions given the features and Model 1’s predictions.
4. https://arxiv.org/pdf/2410.13609 - also focuses on model selection, but under limited labeled data

**Questions:**

Q1. I wasn’t sure how robust the results are; for example, what if you swap the two models you’re comparing? Do you get the same output?

Q2. An additional ablation that would be useful is to evaluate explanations generated using model internals alone (e.g. weights of a logistic regression), without including any predictions on individual examples.

Q3. Is there some guidance on which model should be used for evaluating the quality of the explanation? You mention that you use the same LLM because of “the bias introduced when LLMs process the outputs of the other language models.” — could you provide a citation for this and reason about the implications in your set-up?

Q4. The introduction notes that the models have to be trained on the same dataset; the prompt repeats the same constraint. Do you mean the same task? If not, is there a reason the models need to be trained on the same dataset?

Q5. The manuscript states “While our experiments do not focus on classification tasks” in the related work, but it appears the experiments do focus on classification tasks?

Q6. Certain terms are worth defining; for example, it was not clear what is meant by a “logit lens” in the related work.

Q7. “rather than utilizing the language model head directly, we employ an external LLM to serve as the "model manager," providing a novel means of interpreting and explaining model behaviors.” → could you provide intuition as to why using an external LLM is better than using a language model head directly? Is it because not all models have language model heads, and creating one would be prohibitively expensive?

---

> ### Author Response · Authors · 2024-12-03
>
> Thank you for your thorough feedback. It's clear that you put a lot of care into this review, and we're grateful for your insights and references. Your comments have pushed us to refine our work and think more critically about its limitations and future directions. Below, we address your concerns and questions:
>
> ---
> ### **L1: Models associated with "Model Lake" not chosen**
>
> You’re right that our experiments don’t fully capture the complexity of the “Model Lake” phenomenon. This work was intended as a proof of concept (POC) to explore the feasibility of using LLMs. The verbalizations generated on deep learning models would indeed be more complex, and we're excited about exploring these directions in the future.
>
> ---
> ### **L2: Handling random disagreements and small datasets**
>
> This is an insightful point. We didn’t explicitly examine these cases where disagreements stem from randomness or small dataset variations, but this is a scenario we’d like to explore further. Testing our framework on this scenario would help us understand its limitations better.
>
> ---
> ### **L3: Absence of baselines**
>
> We appreciate your feedback about including baselines, such as training logistic regression to predict Model 2’s outputs from input features and Model 1’s predictions. However, our current focus is on assessing the quality of verbalizations themselves. Comparing this evaluation directly to a logistic regression approach may not align with our objective, as the two methods fundamentally evaluate different aspects of the task.
>
> ---
> ### **L4: Reliance on feature names**
>
> We agree that since the scope of the problem is too large, the current set of experiments in this form cannot be extended to include models that make use of high-dimensional inputs. Relying on feature names does restrict the framework’s applicability to tasks like image classification. This work focuses on structured tabular data as a starting point, but we agree that extending it to high-dimensional inputs (e.g., images) would make it much more generalizable. We hope to explore this in subsequent work with deep learning methods where no feature extraction is needed.
>
> ---
> ### **L5: Changing both LLM_{eval} and LLM_{verb}**
>
> This is a fair critique. In hindsight, holding LLM_{eval} constant would have provided a cleaner evaluation of each verbalizer’s performance. We acknowledge this oversight and will adopt this approach in future experiments.
>
> ---
> ### **L6: Distinct training and verbalization data**
>
> We appreciate your feedback regarding this. We understand that our language may have been a little confusing and that this should have been made clear early on.
>
> ---
>
> ### **Responses to Specific Questions**
>
> **Q1: Robustness to model swapping**
> We didn’t explicitly test swapping the compared models, but it’s a great suggestion. Since we avoided trivial cases (e.g., models predicting all 1s or 0s), we hypothesize the results would hold steady on average. That said, running this experiment is a priority for our future iterations.
>
> **Q2: Evaluations using only model internals**
> Thank you for this insight. Evaluating verbalizations generated purely from model internals (e.g., coefficients or rules) would give us more insights into how well our framework works in this scenario. We plan to expand on this in subsequent works.
>
> **Q3: Which LLM to use for evaluation?**
> Our initial intuition was to evaluate verbalizations using the same LLM that generated them, ensuring alignment in vocabulary and interpretive style. However, we understand that may not be correct. Ideally, verbalizations should be clear and precise enough to be interpreted accurately by any LLM.
>
> **Q4: Why models must be trained on the same dataset**
> The need for the same dataset stems from how we generate and evaluate verbalizations. Our approach compares predictions on the same input instances, which wouldn’t be possible if the datasets differed.
>
> **Q5: Classification focus inconsistency**
> Thank you for catching this inconsistency. We’ve corrected the wording in the revised pdf.
>
> **Q6: Definition of "logit lens"**
> The "logit lens" is a mechanistic interpretability technique used to analyze transformer models. It involves interpreting the model's residual stream—intermediate representations within the model—by projecting them onto the output vocabulary (logits) at various stages of the network. For more information please refer to this [link](https://www.lesswrong.com/posts/AcKRB8wDpdaN6v6ru/interpreting-gpt-the-logit-lens).
>
> **Q7: Why use external LLMs?**
> We chose external LLMs to keep the framework general and adaptable, especially for scenarios where model internals aren’t accessible.
>
> ---
>
> **Examples of verbalization outputs**:
> To help illustrate the process, we’ve added examples of verbalization excerpts in the revised PDF. These examples provide insight into how our approach works and how it could potentially generalize to more complex models.

---

### Official Review · Reviewer_V9RB · 2024-10-31

**Soundness:** 3
**Presentation:** 3
**Contribution:** 2
**Rating:** 3
**Confidence:** 5

**Summary:**

The paper introduces a new task to verbalize the differences in two machine learning models trained on the same dataset. The approach described in the paper uses in-context learning of LLMs to achieve this. The output of LLMs is evaluated by using it to predict one model's output given the output of the other model and the verbalized difference. Experimental evaluation was done on 3 popular classification datasets and 3 popular classification algorithms (Logistic Regression, Decision Trees, and KNN). Two ablation studies aim to understand the effect of including different types of information in the prompt.

**Strengths:**

1. The task is novel and interesting.
2. The evaluation approach and metrics are sound.
3. The paper is well-written and includes enough details of the evaluation.

**Weaknesses:**

1. While the task is described clearly, it is not clear whether it is sufficient to be called a Model Manager.
2. The LLM-based in-context learning approach is not novel as one can imagine any text based task be formulated that way with some reasonable performance.
3. Given the small set of experiments with simple datasets and model families, it is hard to see how this would generalize to more complex real world tasks and models.
4. The code and instructions provided do not seem complete/right. For example, the README says `python lm_manager.py --llm [LM_name] --subject [subject_name]` is the command to run one of the experiments. However neither llm_manager.py nor main.py look aligned with this.

Minor writing issues:
1. spacing near line 166
2. "While our experiments do not focus on classification tasks, we include the feature names to improve interpretability" - confusing as the experiments do seem to focus on classification tasks.

**Questions:**

Could you please include some example verbalization outputs? It is hard to see how this would generalize to complex model families such as XGBoost without the insight into how these outputs look like.

---

> ### Author Response · Authors · 2024-12-03
>
> Thank you for your thoughtful and detailed review as well as for pointing out these issues. We appreciate the time and effort you’ve invested, which has helped us improve the paper. We've addressed some of the points in our revision and provided further clarifications below:
>
> 1. **On sufficiency for being a “Model Manager”**:
> 	We understand and agree with your concerns about the scope of the task. We see this work as the first step in this direction and our goal is to demonstrate the feasibility of using LLMs for verbalizing model differences. Future iterations will focus on scaling this idea to more complex tasks and larger model families to better align with the vision of a comprehensive "Model Manager."
>
> 2. **Novelty of in-context learning**:
> 	While in-context learning itself is not novel, its application to this specific task—verbalizing model differences and evaluating their utility—is, to the best of our knowledge, unexplored.
>
> 3. **Generalizability to complex tasks and models**:
> 	We recognize that our experiments are limited to simpler models and datasets. However, this proof of concept lays the groundwork for future extensions. We are already exploring methods to adapt this framework for deep learning models, which require more sophisticated verbalization strategies.
>
> 4. **Issues with code and instructions**:
> 	We sincerely apologize for the confusion caused by inconsistencies in the README file. The correct command is `python main.py --llm=<LLM> --model=<MODEL_TYPE> --task=<TASK> -- dataset=<DATASET>`. For detailed usage:
>
> ```bash
> usage: main.py [-h] --llm {gpt4,claude,gemini} --model {log_reg,dt,knn} --task {verbalization,generation}
>                [--dataset {blood,car,diabetes}]
>
> Description of your script
>
> options:
>   -h, --help            show this help message and exit
>   --llm {gpt4,claude,gemini}
>                         Specify the LLM to use for evaluation
>   --model {log_reg,dt,knn}
>                         Specify the model type
>   --task {verbalization,generation}
>                         Specify the task to call
>   --dataset {blood,car,diabetes}
>                         Specify the dataset to use
> ```
>
> ##### **Minor Writing Issues**
>
> - We’ve corrected the spacing issue near line 166.
> - Thank you for catching this inconsistency. We’ve corrected the wording in the revised pdf.
>
> ---
> ### **Questions**
>
> **Examples of verbalization outputs**:
> We agree that providing examples is critical to understanding the value of our approach. To help illustrate the process, we’ve added examples of verbalization excerpts in the revised PDF. These examples provide insight into how our approach works and how it could potentially generalize to more complex models.

---

### Official Review · Reviewer_ayuY · 2024-11-02

**Soundness:** 2
**Presentation:** 3
**Contribution:** 2
**Rating:** 3
**Confidence:** 3

**Summary:**

This paper describes an approach to verbalizing the differences in the decision boundaries learned by pairs of machine learning models. The verbalizations are generated by presenting samples of instances and their predictions from the two models, along with some relevant context, to an LLM prompted to describe the differences between the models. The approach is described and then evaluated using three well known tabular datasets and logistic regression, decision tree, and k-NN models. The performance of the system is evaluated using a novel approach that measures the ability of another LLM to generate predictions made using the second model based on the predictions made by the first model and the verbalization of differences between the two models. Based on this evaluation the developed method is shown to perform well with differences between its abilities for different machine learning models explored.

**Strengths:**

The main strengths of the paper are:

- The paper addresses an interesting problem that is becoming more important in the community.
- The methodology proposed is original and promising.
- The proposed approach is clearly presented and described.
- The evaluation methodology is interesting and facilitates large scale evaluation of text generation capabilities.
- Based on the evaluation performed the method performs well.
- Some interesting discussion of the performance of the approach for different kinds of machine leanring models is provided.

**Weaknesses:**

The main weaknesses of the paper are:

- No examples of the verbalizations generated by the approach are provided. This is frustrating for a reader as the value of the approach relies on these being useful to a reader and a small set of examples would easily demonstrate this (or not).

- The motivation of the work is to provide model explanations, but the technique developed generates explanations of differences between models. The authors never explain how this approach achieves the overall aim as a reference model of some kind would always be needed.

- The title is somewhat misleading. Model Manager has a specific meaning in the MLOps community (e.g. SAS Model Manager, Siemens AI Model Manager) which is quite different to what is being presented in this paper. This paper is quite specifically about a technique to generate text explanations of the differences between models and a title more reflective of this would be better.

- The work focuses on explaining non-neural network models (e.g, logistic regression and decision trees) but there is a mismatch with the related work covered, which all focuses on neural network models.

- The presentation and discussion of the results is not clear or consistent - e.g. Why are the Diabetes dataset, Overall Accuracy and k-NNs  missing from Figure 2? Also not clear if the results table (Table 2, 3, and 4) are part of the main body of the paper (referenced as such but then introduced later as additional results).

**Questions:**

As well as addressing the weaknesses described above, it would be useful for the authors to consider the following suggestions and questions:

- "While these methods provide critical insights into individual models and datasets, they do not explicitly dive into verbalizing the differences in model predictions across the feature space. Addressing these limitations and providing interpretable verbalizations is essential for enabling more informed decisions when selecting or developing new and effective models." This statement needs more evidence to support it. How do you know this is essetnial?

- For some references full bibliographic info is not provided. Eg Mu & Andreas 2021 and nostalgebraist 2020.

- It would be worth expanding on whether or F on line 150 is exclusively simple feature names or more detailed feature explanations.

- Not clear what the "verb split" referred to on line 159 is.

- For reproduceability it would be useful to provide more details on how "the datasets were scaled" and the "preprocessing steps" that were applied. Maybe best in an appendix.

- I am not sure that "stratefied" is the correct word to use on Line 276 "we stratified the experiments based on the ..." . Maybe "categorised" would be better?

- Space for more in-depth analysis could be gained by removing the statement of results from the tables shown in Lines 353 - 357 and 374 - 377.

- The reader should be provided with more support to navigate the results. For example "For LR, the inclusion of coefficients results in either performance remaining within the error margin or showing a modest increase (3-5%) across all datasets" is stated but without pointing to a specific table or graph to help the reader understand where this conclusion comes from.

- The following statement in the conclusions "these indicate that the Model Managers can be extended to verbalizing the differences between Deep Neural Networks, especially incorporating approaches that describe the models’ internals (e.g., mechanistic interpretability). " is not well supported by the results presented (which focus on small models and relatively simple datasets).

- The title is somewhat misleading. Model Manager has a specific meaning in the MLOps community (e.g. SAS Model Manager, Siemens AI Model Manager) which is quite different to what is being presented in this paper. This paper is quite specifically about a technique to generate text explanations of the differences between models and a title more reflective of this would be better.

- The presentation and discussion of the results is not clear or consistent - e.g. Why is the Diabetes dataset missing from Figure 2? Overall Accuracy is also missing from this figure? Similarly kNNs are left out of this figure and would be useful to include. There seems to be plenty of space to include all or at least some of these.

- Not clear if the results table (Table 2, 3, and 4) are part of the main body of the paper (referenced as such but then introduced later as additional results). This should be clarified. they are really needed.

---

> ### Author Response · Authors · 2024-12-03
>
> Thank you for your detailed and thoughtful feedback. We appreciate your insights and the time you’ve invested in reviewing our work. They have helped us significantly refine our work. Below, we address your key concerns and suggestions:
>
>
> **1. Examples of verbalizations**
>
> We agree that providing examples is critical to understanding the value of our approach.  To help illustrate the process, we’ve added examples of verbalization excerpts in the revised PDF. These examples provide insight into how our approach works and how it could potentially generalize to more complex models.
>
> **2. Motivation for the work**
>
> Thank you for raising this point. The goal of our work is not to provide generic model explanations but to explicitly differentiate between models by verbalizing their differences. This differentiation is intended to support model selection.
>
> **3. Title clarity**
>
> We acknowledge that the term "Model Manager" may have caused some confusion due to its established meaning in the MLOps community. While our work focuses on verbalizing model differences, the title may indeed be misleading. We will consider revising it to more accurately reflect the scope of our future work. Furthermore, we
>
> **4. On scope for “Model Manager”**
>
> We understand and agree with your concerns about the scope of the task. We see this work as the first step in this direction and our goal is to demonstrate the feasibility of using LLMs for verbalizing model differences. Future iterations will focus on scaling this idea to more complex tasks and larger model families to better align with the vision of a comprehensive "Model Manager."
>
> **5. Results presentation clarity (e.g., Figure 2 and tables)**
>
> We have updated Figure 2 to include the missing Diabetes dataset. Additionally, we’ve corrected our language to clarify that the results in the appendix are not "additional" but represent the complete set of results across all experiments. Thank you for pointing this out.
>
> **6. References**
>
> We’ve updated the references to include complete bibliographic details where they were previously missing (e.g., Mu & Andreas, 2021; nostalgebraist, 2020).
>
> **7. How do we know that providing interpretable verbalizations is essential for enabling more informed decisions?**
>
> We believe this is essential because understanding how models behave relative to each other in different regions of the feature space is critical for informed model selection. The included verbalization examples in the revised PDF further illustrate the value of such insights.
>
> **8. Feature names clarification**
>
> Thanks for pointing this out. The $F$ on line 152 is simply the set of feature names.
>
> **9. Verb split and dataset clarification**
>
> The `verb_split` on line 159 refers to the set of instances used to verbalize differences between models. This is distinct from the training set used to train the models themselves. We’ve clarified this terminology in the revised paper.
>
> **10. Dataset preprocessing**
>
> You’re correct that we should have provided more details on preprocessing. All datasets were normalized and scaled before training so that they could be effectively used with LR and KNNs.

---

### Official Review · Reviewer_7A3j · 2024-11-04

**Soundness:** 1
**Presentation:** 2
**Contribution:** 2
**Rating:** 3
**Confidence:** 4

**Summary:**

The paper presents an LLM based model differencing mechanism. Given two models with (differing) classification outputs, the aim of the LLM is to verbalise the differences in the model outputs. The idea is to analyze the differences in outcomes and provide human-readable text. To evaluate said verbalization, the same LLM is fed the verbalization and asked to predict the output of the second model. If this synthetic label matches the output of the second model, this is considered success. Experiments with standard classifiers are presented and differences artificially induced.

**Strengths:**

The develops an LLM based model manager that could be a viable way to get at model differences in a human understandable way. So the core idea is not without merit.

**Weaknesses:**

A substantive assessment of the weaknesses of the paper. Focus on constructive and actionable insights on how the work could improve towards its stated goals. Be specific, avoid generic remarks. For example, if you believe the contribution lacks novelty, provide references and an explanation as evidence; if you believe experiments are insufficient, explain why and exactly what is missing, etc.

The execution of the key idea is fraught on several fronts. Primarily the choice of the difference levels is entirely unrealistic. Generally, one does not consider models that are this far apart in decision outcomes. I'd expect that the accuracies are also minimally 10 percentage points different (performance measures of the baseline models should be reported as well). A better experiment would be if models differ in the order of a few percentage points - instead of artificially changing models to be so far apart.

While much is said about verbalisation, the paper does not provide a single example of verbal outcomes that are human-reviewable. The paper should do more to describe the verbalisations and perhaps even consider a user study on the efficacy of these to understand model differences.

The evaluation is entirely focused on label outcomes from the verbalisation and $M2$. This is surrogate measure, as there is no guarantee that the verbalisation is reflective of the decision logic. Second by translating to a classification task for the LLM, you induce several artifacts like position bias etc (as you are using the LLM as a classifier). Attributing this to the quality of verbalisation is tenuous.

**Questions:**

Why is Diabetes missing from Figure 2?

---

> ### Author Response · Authors · 2024-12-03
>
> Thank you for your thorough review and constructive feedback. It has helped us improve and refine our work. Below, we address your specific concerns and suggestions in detail:
>
> ---
>
> **1. Unrealistic levels of model difference**
>
> We appreciate and understand your point about the chosen stratification levels. However, it’s important to clarify that these levels are based on the percentage of differing outputs, not performance metrics like accuracy differences. For instance, in one of our experiments using the Blood dataset:
> - **Model 1 Accuracy** = 72.67%
> - **Model 2 Accuracy** = 72.00%
> - **Accuracy difference:** 0.67%
> - **Percentage of differing outputs:** 27.33%
>
> The weights of these LR models are :
> ```
> [[-0.02550171 -0.00254418  0.00384052 -0.0041963 ]]
> [[-0.00686312  0.01219613  0.00417163 -0.00604016]]
> ```
>
> Another example on the Diabetes Dataset
> - **Model 1 Accuracy:** 69.48%
> - **Model 2 Accuracy:** 73.37%
> - **Accuracy Difference:** 3.8%
> - **Percentage of different outputs:** 19.48%
>
> ```
> [[ 0.01104184  0.0068465   0.00088376  0.00107555 -0.00154701  0.03248901
>    0.00421504  0.00321098]]
> [[ 0.00447317  0.00560759 -0.00181089  0.00081074 -0.00072719  0.00588984
>    0.00260115  0.0028648 ]]
> ```
>
> This highlights that while the accuracy difference appears minimal, the output divergence can still reach significant levels. By focusing on such stratifications, we aim to ensure a nuanced comparison of the models’ behavior.
>
> ---
> **2. Lack of example verbalizations**
>
> We acknowledge that the absence of concrete verbalization examples may have caused confusion. To address this, we have now included verbalization excerpts in the revised version of the paper (e.g., Table 2 and Table 3). These examples illustrate the nature of the differences captured by the framework and enhance the transparency of our results.
>
> ---
> **3. Missing Diabetes dataset in Figure 2**
>
> Thank you for catching this oversight. We have corrected Figure 2 in the revised PDF to include results from the Diabetes dataset.
>
> ---
> **4. Verbalizations reflecting decision logic**
>
> We understand and agree with your concern regarding the interpretability of verbalizations. While the accuracy of the evaluation (80-90% in some cases) supports the validity of some of the verbalizations, we recognize that this does not guarantee complete alignment with the models’ decision logic. As a first step, our framework provides a promising direction, and we aim to refine the pipeline to reduce artifacts (e.g., position bias) and improve evaluation robustness in future work.

---

### Meta-Review · Area_Chair_dbnf · 2024-12-16

**Metareview:**

After reading the reviewers' comments and reviewing the paper, we regret to recommend rejection.

Taking from reviewer ayuY: The paper describes an approach to verbalizing the differences in the decision boundaries learned by pairs of machine learning models. The verbalizations are generated by presenting samples of instances and their predictions from the two models, along with some relevant context, to an LLM prompted to describe the differences between the models. The paper presents some interesting ideas and that are original and may be promising, however that work fails short of clearly articulating the practical use of the results, and the description of the results is not clear and consistent. Hence, a reader may find it difficult to grasp the key conclusions and advantages of the proposed approach.

I agree with this view of the paper.
The authors may consider to further clarify the practical use cases, and better streamline their results.
In addition, the conclusions should be better articulated to highlight the key advantages of this work, compared to existing literature and competing material.

**Additional Comments On Reviewer Discussion:**

The authors have been proactive in addressing the comments raised by the reviewers, and reviewer KAiZ was engaged responding to the authors.

We agree with the reviewers comments, and recommendations, noting some of the weaknesses that we believe may remain and mentioned in the metareview.

No ethics review raised by the reviewers, and we agree with them.

---

### Decision · Program_Chairs · 2025-01-22

Reject